# Optimal Duration of In-Vehicle Data Recorder Monitoring to Assess Bus Driver Behavior

**DOI:** 10.3390/s23218887

**Published:** 2023-11-01

**Authors:** Rachel Shichrur, Navah Z. Ratzon

**Affiliations:** 1Occupational Therapy Department, Ariel University, Ariel 4077603, Israel; 2Department of Occupational Therapy, Tel Aviv University, Tel Aviv 6997801, Israel; navah@tauex.tau.ac.il

**Keywords:** professional drivers, naturalistic driving, G-sensor, advanced driver assistance system, vision-based technology

## Abstract

This study examined the optimal sampling durations for in-vehicle data recorder (IVDR) data analysis, focusing on professional bus drivers. Vision-based technology (VBT) from Mobileye Inc. is an emerging technology for monitoring driver behavior and enhancing safety in advanced driver assistance systems (ADASs) and autonomous driving. VBT detects hazardous driving events by assessing distances to vehicles. This naturalistic study of 77 male bus drivers aimed to determine the optimal duration for monitoring professional bus driving patterns and the stabilization point in risky driving events over time using VBT and G-sensor-equipped buses. Of the initial cohort, 61 drivers’ VBT data and 66 drivers’ G-sensor data were suitable for analysis. Findings indicated that achieving a stable driving pattern required approximately 130 h of VBT data and 170 h of G-sensor data with an expected 10% error rate. Deviating downward from these durations led to higher error rates or unreliable data. The study found that VBT and G-sensor data are both valuable tools for driving assessment. Moreover, it underscored the effective application of VBT technology in driving behavior analysis as a way of assessing interventions and refining autonomous vehicle algorithms. These results provide practical recommendations for IVDR researchers, stressing the importance of adequate monitoring durations for reliable and accurate outcomes.

## 1. Introduction

The collection of accurate, reliable, and valid driving information over a wide range of conditions, including the occurrence of undesirable events, has been made possible by monitoring technologies such as the sensor-based in-vehicle data recorder (IVDR). An IVDR is a comprehensive data resource for gaining insights into real-world driving behavior by observing drivers in their natural, everyday environments. The IVDR has been found to be an effective assessment tool for identifying drivers’ behavior patterns and characteristics, including the propensity to indulge in risky driving behaviors [1,2,3]. Using an IVDR can enhance driver safety by reducing collisions, near-miss incidents, injuries, emissions, and fuel use, thus achieving sustainability goals. Using IVDRs in vehicles can help drivers maintain a steady velocity and avoid the unnecessary acceleration and braking to which drivers frequently subject their vehicles. This enhances overall driver safety [4].

### 1.1. Data Sampling Duration

Unlike traditional laboratory-based experiments or simulations, where participants perform in controlled settings, naturalistic studies involve collecting data from participants as they go about their normal driving routines [1,5]. Naturalistic driving studies typically collect data over extended periods ranging from many days to multiple months and even years, providing a longitudinal perspective on driving behavior [6]. This allows researchers to identify trends and patterns that would not be apparent in shorter studies. However, there is a lack of consensus in the literature about the optimal practical duration of data “sampling” required to establish an accurate driving profile. Suggested IVDR sampling amounts vary widely across studies, ranging from 80 h [7] to 400 h [8] to 2200 h [9] per driver. These different approaches toward the optimal monitoring duration can lead, on the one hand, to short monitoring periods that may not capture representative behavior. On the other hand, overly long periods may lead to a loss of interest among participants and concerns about loss of privacy.

The need to determine the proper measurement duration also emerged in other aspects of naturalistic driving studies. For example, Shangguan et al. (2021) dealt with the lack of precision in predicting real-time driving risk due to the variations in the timing of observations and predictions, as the time periods used in current research about driving risk status prediction vary widely, from just a few seconds to quite a long time [3].

The limited evidence available on this topic and the wide range of sampling durations suggested by the existing studies highlight a crucial gap in understanding the best monitoring duration for accurate driving assessments. Inadequate monitoring can result in unreliable assessments, either underestimating or overestimating risky driving behavior.

### 1.2. Driving Behavior—Factors Affecting Optimal Sampling Time Frame

In order to determine the minimum required duration for accurate assessing of driving behavior, a few factors must be considered, including the type of monitoring technology that is utilized. Most of the literature on naturalistic driving and IVDR systems has focused on using GPSs, accelerometers, and on-board diagnostics [10,11]; doing so evaluates driving behavior only through various vehicle maneuvers such as hard brakes, sharp turns, strong accelerations, and speed (the G-sensor, a three-axis accelerometer). G-sensor technology does not take into consideration what is occurring in the environment around the vehicle (such as the distance from other vehicles and pedestrians on the road). However, despite the critical importance of the environment in the comprehensive understanding of driving compounds [12], many naturalistic driving studies have ignored exogenous factors and measured only the interaction between the person and the vehicle [10].

### 1.3. Driving Safety—Driver–Vehicle–Environment Interaction

Driving risk is related to various traffic factors, including behavioral characteristics, vehicle characteristics, and driving environment characteristics [13]. Lindstrom-Forneri et al. [12] based their Driving as an Everyday Competence (DEC) model, a comprehensive theoretical model for understanding the full range of factors that enable or limit safe driving, on the dynamic interaction between individual and environment. This approach led the current study to examine vision-based technology (VBT), an emerging technology in the field of advanced driver assistance systems (ADASs), increasingly used in transportation applications and autonomous driving technology to monitor drivers’ behavior and enhance driver safety. VBT focuses on monitoring the relationships within the driver–vehicle–environment complex by observing the vehicle’s surroundings, detecting potential hazards, real-world driving behavior, and road conditions.

Building on the statistical analysis method utilized in our previous study of G-sensor technology [14], the current study focuses on the capabilities and limitations of VBT. VBT can be a basis for supporting vehicle autonomy [15]. Manufactured by Mobileye®, the VBT system performs real-time scene interpretation, analyzing different objects on the road, classifying them, and determining if action is needed. These environmental factors are integrated into the measured data as indicators that serve to assess the risk of a car crash [16].

## 2. Literature Review

### 2.1. Safety Concerns in Bus Transportation

About one-third of the world’s traffic crashes involve professional drivers [17,18,19]. Useche et al. [20] provide a detailed description of the high risks of passenger transporters, such as taxi and bus drivers, highlighting the relationship between psychosocial risk factors at work (high performance demands, low perceptions of task-control, minimal social support, effort/reward imbalances) and adverse road safety outcomes such as work-related traffic accidents, traffic violations, and risky driving behavior [21,22]. Professional drivers often experience stressful working conditions due to long work hours, shift work, traffic congestion, and difficult interactions with passengers [23,24,25]. Thus, an enhanced characterization of unsafe driving behaviors is expected to contribute greatly to road safety research.

### 2.2. Evolution of Sensors

The rise of autonomous and semiautonomous vehicles has driven the development of advanced sensors, including light detection and ranging (LiDAR), radar, and computer vision systems, which are all vital for autonomous navigation, collision avoidance, and environmental awareness [26]. This evolution in sensor technology aligns with broader automotive trends, such as increased connectivity, automation, data analytics, and environmental awareness; it significantly enhances vehicle safety, performance, and efficiency while providing valuable data for vehicle manufacturers, researchers, and users [3]. Furthermore, the instrumentation of vehicles has become increasingly miniaturized and unobtrusive; the ability to inconspicuously install measurement devices within a personal vehicle is due to recent advances in the miniaturization of sensors, data storage devices, communication systems, and video technologies [27].

### 2.3. Role of NDS and VBT in Addressing Human Factors-Related Safety Concerns

In a rapidly evolving world of advanced vehicle safety technologies, comprehensive innovations in driver state sensing, vehicle autonomy, and a range of ADAS capabilities, human factors remain a crucial component of driving [28]. Indeed, research shows that most traffic accidents are associated with unsafe driving behaviors [13,29]. Although many studies have attempted to evaluate risk-driving behavior [30,31], there is a notable shortage of research that has investigated the factors underlying individual driving behavior and quantifying the driving behavior patterns of professional drivers via long-term naturalistic driving data. The quantitative estimation of driving performance and behaviors using VBT is likely to be helpful in measuring a driver’s risky driving behavior to decrease traffic accidents.

The advent of new technologies that enable novel warning systems requires extensive testing to demonstrate that they result in a reduction in driving risks [32]. The validation of these technologies requires statistical analysis of real data about risky situations, including near-misses and accidents [32]. VBT consists of several technologies identified by experts as having the potential to significantly impact road safety [26]. As an ADAS, the VBT provides a means to proactively reduce the rate of accidents. It can warn car drivers about multiple potential crash hazards (e.g., forward collisions, running off the road, or overly aggressive driving) by identifying them before they escalate into actual accidents.

The aim of the present study is to examine the optimal sampling time frame required to reach a stable driving pattern (event rate) among professional bus drivers, utilizing two IVDR systems: (a) G-sensor-based IVDR technology and (b) a VBT, which adds the evaluation of environmental factors to the monitoring data.

## 3. Methodology

### 3.1. Participants

The study was approved by the Helsinki ethics committee (RMC-0103-10). A random convenience sample of 77 male bus drivers was recruited from a large urban bus company in Israel. All participants gave their written informed consent after the nature and purpose of the study, including the installation of the IVDR (without feedback) in their vehicle, were described. They were also informed that the information collected by the system would not be disclosed to their employer. These precautions were designed to minimize the effect of the instrumentation on the drivers’ behavior. The participants were given a moderate honorarium for their time and received traveling expenses related to study participation.

The recruitment of drivers and the installation of instruments in the buses were carried out gradually over two years. Only male bus drivers were recruited due to the very small number of female drivers in this bus company. Sixteen drivers were ultimately excluded due to insufficient VBT data (*n* = 61); 11 of those 16 drivers also had insufficient G-sensor data (*n* = 66). Drivers were 27–69 years old (M = 52.3, SD = 9.3). The cohort’s mean time for possessing a driver’s license was 32.3 years (range = 12–50, SD = 9.7), and the mean working time as a professional bus driver was 20.9 years (range = 1–45, SD = 12.7).

Public transportation is provided by the bus company seven days a week and 24 h a day. The standard work week of a company employee is six days, with shifts lasting for 8 h (i.e., day, evening, or night shifts). Most study participants worked split shifts and overtime, such that their on-the-job time ranged from 9 to 11 h per day and 54 to 66 driving hours per week. They drove mainly in an urban area, encountering a range of levels of difficulty characteristic of this country’s driving routes (e.g., busy city streets vs. less-trafficked roads). Drivers worked in most weather conditions and seasons (but did not encounter snow or ice).

### 3.2. IVDR Monitoring

The IVDR monitoring methods utilized in the current study consisted of two separate systems: a G-sensor and a VBT. The G-sensor is the vehicular unit (VU), a small (10 × 5.5 × 2 cm) box installed in a protected place in the vehicle dashboard. The system establishes and maintains a direct connection with the controller area network (CANBUS) and the on-board diagnostics (OBD) system. The VU system allows for measurements of the vehicle’s G-force with a 3-axis accelerometer. It records detailed information about the vehicle’s position, movement, and vertical and horizontal acceleration. Trip information is stored during the driving period (start and end times, locations, distance, and duration), and the type of undesirable events that occur during the trip is also stored (see Section 3.4). The G-sensor evaluated in a previous study [14] was developed by Traffilog LTD, while the G-sensor used in the present study was manufactured by ISR Corp (www.isrfleettrack.com (accessed on 1 January 2020)).

The second IVDR system, VBT (C2-270™), is a car sensor about the size of a toll tag mounted on the inside of the vehicle’s front windshield that uses a high-resolution single-lens camera and a visual display. The VBT system continuously measures the distance and relative speeds of objects on the road, predicting their path and calculating the risk of the vehicle colliding with them (Table 1). Developed in 1999 by Prof. Amnon Shashua (http://www.mobileye.com/ (accessed on 30 March 2021)), the VBT is an ADAS using motion detection algorithms and artificial vision technology that functions as a “third eye” on the road [33]. It can identify objects within the forward view of the camera that may pose a threat to the vehicle, such as other vehicles, bicycles, motorcycles, and pedestrians, in both daytime and nighttime conditions [34]. The VBT possesses a forward collision warning component that detects whether a crash is imminent by computing the “time to contact” (TTC), considering the host vehicle’s speed, relative speed, and relative acceleration. The last two are calculated from the change in the image size of the target (scale change); in the case of a predicted crash, a “critical warning” is issued to the driver, soliciting a corrective response. Although a VBT can provide drivers with real-time audio and visual warning alerts, the present study utilized the recording device in a “no feedback” mode in order to focus on its ability to assess drivers’ risk levels without intervention.

### 3.3. Procedures

VBT IVDRs were installed by the bus company technicians in the participants’ vehicles, after which detailed information about unsafe events that occurred during driving was recorded. The buses were monitored during the participants’ working shifts while driving on their regular urban routes. To ensure that the participant was the only driver of the vehicle, each driver entered their personal code before driving and all data recorded were synchronized with both vehicle-specific and driver-specific codes. The IVDR data included all weather types and seasons, day and night. The maximum driving duration of the drivers in the study was up to 2855 h for those with a G-sensor and up to 1700 h for those with a VBT. We note two factors that limited a complete data set. First, although the VBT IVDR systems were installed in the buses assigned to all 77 participating drivers, the company occasionally required that some drivers drive a bus that was not equipped with a monitoring device. This resulted in missing data. Second, technical problems were encountered as a result of improper installation. Thus, IVDR data for 21–23% of drivers in each of the IVDR event categories were lacking. Therefore, 16 drivers without complete IVDR monitoring data were excluded from the relevant data analyses.

### 3.4. Data Collection

IVDRs installed in the vehicle of each participant recorded detailed information about the type of undesirable events that occurred during driving. Events were recorded as the number of occurrences per hour.

#### 3.4.1. Raw Data

The raw data of events recorded by the G-sensor and VBT were stored in two separate data files with similar formats. Each file line represents a single event in time, with its appropriate value: one of three types of G-sensor events (characterized by *x*-, *y*-, and *z*-axis values) or one of four types of VBT events. The raw data derived from the G-sensor’s 3D linear accelerometer (m/s^2^) were divided into three types of events:(a)Acceleration.(b)Braking (deceleration).(c)Left and right turns.

Based on recommendations from previous studies [35,36], predefined threshold G-sensor values were filtered to retain only events within the range of 0.4–2 g (in absolute values). The VBT data were classified into events: (a) forward collision monitoring (FCM), (b) urban forward collision monitoring (UFCM), (c) unsafe headway monitoring (HM), or (d) sudden lane deviations (LDs). Detailed definitions are shown in Table 1.

In the present study, each G-sensor or VBT event was considered as an absolute value, regardless of its severity, and only the total mean event rate per hour was used for calculations. Repeating or closely adjacent events were removed to avoid recording the same event twice. For example, a sharp turn followed by a strong acceleration was considered as the same event if the two occurred less than 1 min apart.

Data analysis began by outlining the statistics of drivers’ undesirable event data in order to understand the frequency of dangerous events per hour among the bus drivers. Next, the IVDR data were analyzed using the calculation method of the expected error, in order to find a reasonable time frame for reaching a stable driving pattern.

#### 3.4.2. Computing the Mean Event Rate per Hour

To compute the undesirable event rate per driver:Each G-sensor and VBT data file was sorted according to their timestamp in ascending order to create two new vectors, representing the time difference between timestamps;All time differences greater than 2 h were set to zero (as these were generally found to represent intervals when drivers took a break or finished their workday);A chronological matrix was created using cumulative sums, with each column representing a different type of event and each row a sampled hour;Similar to point 3, we created a vector with the total number of events per hour;A mean event rate per hour was calculated by averaging each column as a function of time (*T*′).

Following steps 1–5, for each time (limiting the calculation to a certain time *T*), the relative error per driver was computed (i.e., the mean over the entire duration of the driver’s performance was sampled) and iterated on *T* (increasing by one hour in each iteration). The total mean event rate is represented by this formula:$$\frac{1}{T\prime }\sum_{i=1}^{T\prime }(a_{i}+b_{i}+c_{i})$$
where, *a_i_*, *b_i_*, and *c_i_* represent G-sensor events or VBT events. The rate of undesirable events per hour was defined as R. For each driver, the mean event rate R′ was computed by dividing the total number of events by the total hours driven (a high rate indicating dangerous driving); the underlying assumption was that a sufficient sampling duration of driving hours leads to a more stable statistical estimate of R. Total driving time was defined as *T* hours, and the mean event rate for each time between 0 and *T* was defined as *t* and computed as follows: for driving time *t* (0 < *t* < *T*), the mean event rate up to time *t* was the number of events up to time *t* divided by *t*: R′(*t*) = #{events occurring up to time *t*}/*t*. In this notation, the eventual event rate for each driver R′ = R′(*T*) is equal to the mean event rate over the entire driving time *T*, treating the rate as a function of time.

Considerable differences between participants produced unstable driving patterns over longer periods of time, which led to a fairly high error rate. Therefore, to reduce the variance, instead of using the mean event rate per hour, the mean square root (sqrt) of the event rate per hour was computed.

#### 3.4.3. Computing the Mean Square Root (Sqrt) of the Event Rate per Hour

In order to compute the mean square root (sqrt) of the event rate per hour:Similar to steps 3–4 above, a vector was created from the square root of the total number of events per hour;The mean square root of events per hour was calculated by averaging the column computed in the first step.

The formula represents the mean sqrt of the event rate:(1)$$\frac{1}{T\prime }\sum_{i=1}^{T\prime }\sqrt{a_{i}+b_{i}+c_{i}}$$

Therefore, the “VBT total risk score” in this study refers to the square root of the total VBT events per hour and the “G total risk score” refers to the square root of the total G-sensor events per hour.

#### 3.4.4. Expected Error

The expected error is the percent error of the mean sqrt event rate that would be obtained if measuring was stopped at a certain timepoint (i.e., truncation time) compared to measuring the total event rate up until the end of all the hours driven. The expected error was calculated by deducting the mean event rate produced with a truncated sample from the mean event rate over the entire driving time (taking the absolute value of that difference), divided by the mean event rate across the entire driving period. The decision to select a 10% threshold limit for the expected error rate was based on the findings presented in “Working paper 5” of the Traffic Conflict Studies Report from the National Cooperative Highway Research Program [37].

## 4. Results

Table 2 shows descriptive statistics for undesirable G-sensor events: (a) turns per hour, (b) accelerations per hour, (c) braking per hour, and (d) sum of all G-sensor events per hour. Table 3 presents descriptive statistics for undesirable VBT events: (a) FCM, (b) UFCM, (c) HM, and (d) LDs.

As shown in Table 2 and Table 3, the frequencies of all undesirable events (G-sensor and VBT) and their standard deviations are relatively high among bus drivers. The mean number of VBT events per hour is about four times greater than the mean number of G-sensor events per hour. Two stages of analysis were conducted to find the reasonable sampling time for IVDR measurement to analyze driving patterns and changes in the behavior of drivers over time.

### 4.1. Stage 1: Searching for Driving Patterns

Using the information collected with the IVDR systems, the mean event rate per hour from the G-sensor and VBT was computed for each of the 66 or 61 participants, respectively, as shown in Figure 1 and Figure 2.

Figure 1 shows that most participants (70%) drove for up to 250 h and had a range of 2.25–12.3 events per hour (left, blue rectangle). A small group of five drivers drove for only 15 to 110 h and had the highest mean event rate of 15–22 events per hour (red oval). The 25% of participants who drove 400 to 1500 h had a range of 3.6–15 events per hour (large purple rectangle).

Figure 2 shows that most participants (83%) drove for less than a total of 250 h and had a high mean event rate of up to 18.5 events per hour (left blue rectangle). The same small group of five participants from Figure 1 drove for a total of 15 to 135 h and had a high range of 22–34 events per hour (red oval). Another small group of five participants drove for a total of 710 to 970 h and had a range of 6–18.5 events per hour (right green square). Another three participants drove for 1300 to 1400 h and had a low mean event rate of 6.6–12 events per hour (right dotted circle).

To demonstrate the differences between driving patterns among the study drivers, and to show that different rates of data stabilization should be considered, mean G-sensor and VBT event rates are shown in Figure 3 and Figure 4, respectively, with each line representing three drivers who have different driving patterns (Driver A, Driver B, and Driver C).

There are several similarities between the typical driving patterns of each driver. In Figure 3, Driver A’s mean event rate is the highest, showing a steeply increasing rate, until it stabilizes at around 70 h with a mean of about 14 events per hour. Driver B is representative of the majority of drivers participating in this study, accruing around 6.5 events per hour during the first 15 h of sampling, then experiencing a declining and eventually stable mean event rate of 3.5 events per hour starting at around 50 h of driving. Similar effects can be observed for Driver C, but on a smaller scale; the rate stabilizes at around 0.5 events per hour within the first hours of driving, a rather low mean rate relative to the sample as a whole.

In Figure 4, as in Figure 3, Driver A’s mean event rate is the highest, showing a steeply increasing rate that begins to stabilize at around 15 h, with a mean of about 24 events per hour. At around 100 h, the event rate begins to gradually decline, until it stabilizes at a mean rate of about 19 events per hour. Similar patterns can be observed for Drivers B and C, but on a smaller scale. The event rate of Driver B stabilizes at around 75 h, at a mean of 10 events per hour, and the rate of Driver C stabilizes at around 20 h, at a mean of six events per hour.

These differences in the time it takes for the event rate to stabilize and the differences in the mean event rates per hour of driving raise the question of how much monitoring time is needed for the event rate, on average, to stabilize and thereby demonstrate reliable driving patterns.

### 4.2. Stage 2: Finding the Minimal Sampling Duration Needed to Obtain a Stable Driving Pattern

The graphs in Figure 5 and Figure 6 demonstrate two methods for calculating the minimal sampling duration of driving monitoring, including the range of percentages in expected error. One method is to calculate the expected error of the mean event rate, and the second is to calculate the expected error of the square root of the mean event rate. These two methods are depicted in Figure 5 for the G-sensor data and in Figure 6 for the VBT data.

The *x*-axis represents the total number of hours of monitored driving. The right *y*-axis represents the number of drivers in the sample at each number of driving hours. The number of drivers was not the same across all durations of driving hours since the number of drivers remaining in the sample decreased as the number of driving hours increased. The left *y*-axis represents the expected error of the estimated event rate for the number of participants driving at each of the specified numbers of hours. The two horizontal lines represent the range in expected error, between 10% and 20%.

According to Figure 5, when the sample of the G-sensor event rate (the dotted line) is truncated at x = 100 h, the expected error of the event rate is ~21%, i.e., relatively high and less reliable. Even when the sample is truncated at x = 300 h, the expected error is ~15.4%, still higher than desired. To reach an expected error of precisely 10%, a collection of ~500 h of G-sensor data is needed.

When the sample of the sqrt G-sensor event rate (dashed line) is truncated at x = 100 h, the expected error is ~11.4%. Truncating the sample at x = 50 h results in a reasonable expected error of 13%. However, to reach an expected error of 10%, a collection of 170 h of driving data is needed. At x = 300 h, the expected error is about 8%.

As shown in Figure 6, when the sample of VBT event rate (dotted line) is truncated at x = 100 h, the expected error is ~26%. i.e., relatively high and insufficiently reliable. When the sample is truncated at x = 300 h, the expected error comes closer to the desired 10% (~11.1%). To reach an expected error of precisely 10%, a collection of ~500 h of driving data is needed.

When the sample of the sqrt VBT event rate (dashed line) is truncated at x = 100 h, the expected error is ~12%. Truncating the sample at x = 50 h results in a reasonable expected error of 12.3%. However, to reach an expected error of 10%, a collection of 130 h of driving data is needed. If x = 300 h, the expected error drops to about 6%.

## 5. Discussion

Given the ongoing lack of consensus in the literature, this study aimed to establish an optimal monitoring duration to correctly detect and assess changes in professional bus driving patterns over time while achieving a balance between accuracy and efficiency in data collection efforts. Indeed, recordings from both G-sensor technologies and VBTs demonstrated significant differences in the driving patterns of professional bus drivers (mean event rate per hour of driving and time taken for event rate to stabilize). Although both effectively achieved a 10% error rate, the VBT required a shorter monitoring duration (130 h) compared to the G-sensor (170 h); decreasing these durations led to higher error rates or unreliable data. These results provide valuable data to support the selection of alternate methods of recording driving patterns. Together, these findings highlight the feasibility of reliably monitoring the time needed to establish valid mean event rates and stable driving patterns. Too short a duration may lead to missed changes, wasting resources. Conversely, overly long monitoring, like excessive statistical power, may yield findings with little real-world practical significance.

The results identified different pattern types regarding the number of undesirable events and the variable time needed to stabilize them. These findings align with other research that investigates driving behaviors, such as Useche et al.’s [20] work that highlighted the crucial role of driving styles in moderating the association between job strain and occupational traffic crashes among professional drivers, particularly among those with “reckless and careless”, “anxious”, and “angry and hostile” driving styles. Their results emphasized the practical implications of understanding the role of driving styles when developing evidence-based interventions to enhance road safety for professional drivers and reduce occupational traffic incidents at both the organizational and individual levels [20]. Additional research regarding the relationship between the event rate and individual psychological factors is needed.

Although the use of kinematic data (G-sensor) and vision-based data (VBT) during real-time driving was previously explored by Musicant et al. [9] to document driver behavior, their sample size was limited (only 10 drivers) and they experienced technical challenges due to large differences in the number and type of events recorded. Nevertheless, they recognized the importance of this approach to achieve a more comprehensive understanding of driver behavior by combining kinematic and vision-based driving event data. In a 2014 [14] study of cab drivers’ IVDR data using G-sensor technology, we found that 300 driving hours was the minimal time frame for reaching a stable driving pattern with an expected error rate of 10%. The same study showed that sampling fewer than 100 driving hours resulted in an expected error of 25%, which is not acceptable when assessing driver event rates. The drivers tested in the 2014 (cab drivers) and current (bus drivers) studies differed, although both were exposed to similar human factors issues (e.g., long work shifts, high stress) that affected their driving ability and increased their risk of involvement in road accidents [20,38,39].

Variations in the type of vehicle driven [40] and the height of the embedded technology likely partially explain the results in the [14] study. Indeed, the potential danger associated with vehicles of large size and weight, higher centers of gravity, and other unique characteristics was discussed by Zhou and Zhang [41]. Sedans of the type used by cab drivers have a much lower physical profile than a typical bus; despite IVDR placement in exactly the same position on the dashboard of a bus or a cab, differences in the height of the G-sensor, reflected most notably in turns and side fluctuations due to road anomalies (e.g., bumps, holes), can affect the collected data. Therefore, when determining optimal driver data sampling times, one must also take into consideration the driver’s vehicle type and size. This is particularly noteworthy, as our study revealed a six-fold increase in the frequency of all undesirable events recorded by the G-sensor, as well as greater variability among bus drivers compared to what had been observed in cab drivers [14]. Differences between bus and cab drivers’ driving routes (predetermined subsets of streets for bus drivers versus more varied routes for cab drivers) may also impact the level of exposure to accident risk.

Given the differences between the two professional driver experiences, the present study attempted to identify the minimal duration of monitoring at which the rate of undesirable driving events for bus drivers stabilized by using a computation of the expected error for different sampling durations. Calculating an expected error rate of 10% according to the number of undesirable events per hour corresponded to a relatively high number of driving hours, requiring the collection of 500 h of either G-sensor or VBT data for each bus driver. Analysis of the bus drivers’ data showed that the cause for the slower stabilization of the data was a large variance in the mean event rates per hour of driving, among bus drivers. Therefore, the square root of the number of events was utilized to reduce the variance, making data collection over fewer hours, and with lower error rates, possible.

The main study finding demonstrated that, by using the square root of the number of undesirable events, it was possible to obtain a stable driving pattern for bus drivers (i.e., expected error of 10%) when collecting only 170 driving hours of G-sensor data and 130 driving hours of VBT data—a much more practical, feasible sampling duration than the 500 h identified in the first round of data analysis. Indeed, these reduced sampling durations translate into approximately 3 weeks of G-sensor bus driving data and approximately 2 weeks of VBT bus driving data, when excluding rest days. In contrast, sampling durations of fewer or more hours were either inadequate or excessive. Sampling less than 50 driving hours did not result in a reliable measure for assessing the drivers’ event rate (G-sensor expected error = 13%; VBT expected error = 12.4%). Sampling 50–170 h of G-sensor data or 50–130 h of VBT data resulted in a stable measure but was less recommended, as it increased the expected error beyond the threshold limit of 10% [37]. Assessing more than 170 G-sensor driving hours or more than 130 VBT hours lowered the expected error, but the decrease in the marginal (per-unit) output of the process made the assessment less effective. 

The current study concludes that the VBT is a more effective assessment tool for identifying drivers’ behavior patterns and characteristics within a reasonable time frame. Support for adopting this technology had been previously reinforced by the finding that the VBT total risk score was the only significant contributing factor to predicting crash risk; for each point of increase, the odds of being involved in a crash increased by a factor of 1.55 [16]. The environmental advantages of vision-based technologies over relying solely on GPS are well known in the literature. In specific situations where GPS usage is challenging, such as during adverse weather conditions or indoors with a weak signal, the utilization of a vision-based system allows for effective navigation by analyzing the visual characteristics of its surrounding environment, thus overcoming these issues [15]. Given that vision-based technology is a fundamental component of the sensory suite in autonomous vehicles [15], it can support drivers in making informed safety-promoting decisions. As technology continues to advance, vision-based systems are expected to play an increasingly vital role in the widespread adoption of autonomous vehicles.

## 6. Limitations, Conclusions, and Future Studies

This empirical study demonstrated how a standardized, naturalistic study can help stakeholders employ IVDRs and VBT to achieve safer, efficient transportation. It addressed how the duration of data collection impacts the reliability of driver behavior assessment. The study presented findings on the stabilization of driving patterns and highlighted the ability to achieve accurate conclusions about driver behavior when the monitoring duration is sufficiently long. Specifically, the results suggest that an optimal sampling time frame appears to be required for achieving a stable and reliable driving pattern (event rate) for VBT (approximately 130 h) and G-sensors (around 170 h). Sampling less than this amount resulted in either higher-than-recommended error rates or unreliable data.

Since a homogeneous group of participants (mainly urban bus drivers) was tested in this study, the generalizability of its findings is somewhat limited. Future studies should replicate the results among other driver populations after conducting a small-scale pilot study to determine a reasonable sampling time frame for this population using the methods described here. Despite differences between driver populations and their vehicles, comparing similar kinds of driver populations to one another, such as cab and bus drivers, is critical to achieving valid interpretation and generalization of IVDR data.

The present study focused on determining the sampling time frame for IVDR measurements without considering the relationship between driving duration and rest time, which may have affected the event rate. Additionally, variations in IVDR products from different manufacturers (including differences in the thresholds of undesirable events and event severity levels) may explain some of the variance in the mean event rates across drivers. The present study did not address the severity of events and differences between conditions and scenarios, regardless of the potential importance of such factors. Moreover, the required sample number may differ depending on modeling approaches [13]. Some nonparametric methods may not need more samples to produce a reliable result compared to parametric models. Therefore, findings in which sample size uses only error terms may not be useful for applying various modeling methods and approaches [9]. Future studies should address all of these limitations.

There were several limitations related to the VBT used in this study. Several VBT events, such as sudden lane deviations (LDs) and unsafe headway monitoring (HM), are not active at low speeds, although this information is crucial for understanding bus drivers’ behavior when entering and departing the bus station or when approaching an intersection. In addition, pedestrian collision warning (PCW) is not active at night. Finally, the camera’s field of view overlaps with the range of the car’s forward direction of travel; therefore, potential hazards beyond the camera’s view are not captured. This limitation has been solved in autonomous vehicles by placing eight cameras on all sides to achieve a 360-degree field of view.

In conclusion, the study findings offer insights for researchers working with advanced IVDR technologies to assess changes over time among professional bus drivers, possibly leading to more consistent measurement practices. Determining the optimal monitoring duration enhances the identification of risky driving behaviors and a reduction in accidents and costs. Guidelines such as those resulting from this study can improve research validity and foster meaningful conclusions and interventions.

## Figures and Tables

**Figure 1 sensors-23-08887-f001:**
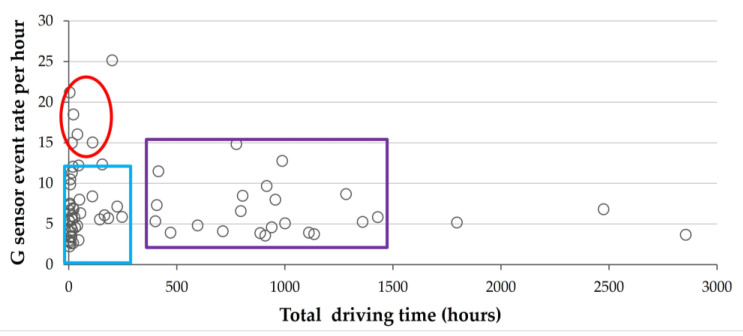
Mean G-sensor event rate per driving hour for 66 drivers. 70% drove ≤ 250 h with a 2.25–12.3 events/hour rate (narrow blue rectangle). A subgroup (*n* = 5) drove 15–110 h, averaging 15–22 events/hour rate (red oval). 25% drove 400–1500 h, with a 3.6–15 events/hour rate (purple rectangle).

**Figure 2 sensors-23-08887-f002:**
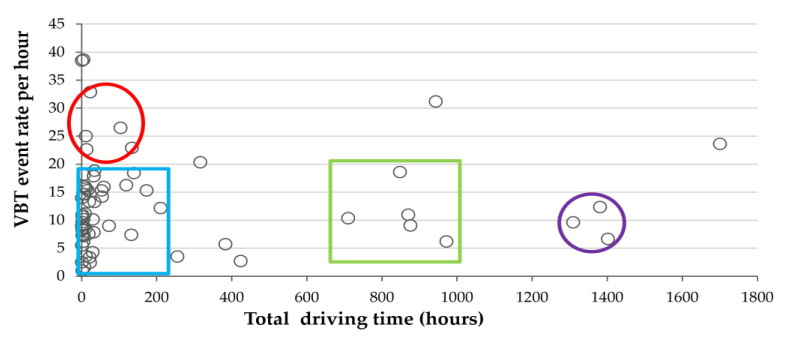
Mean VBT event rate per driving hour for 61 drivers. 83% drove < 250 h with up to 18.5 events/hour rate (narrow blue rectangle). Subgroup (*n* = 5) from Figure 1 drove 15–135 h with 22–34 events/hour rate (red oval). Another subgroup (*n* = 5) drove 710–970 h with a 6–18.5 events/hour rate (green square). Three participants drove 1300–1400 h, averaging 6.6–12 events/hour rate (purple circle).

**Figure 3 sensors-23-08887-f003:**
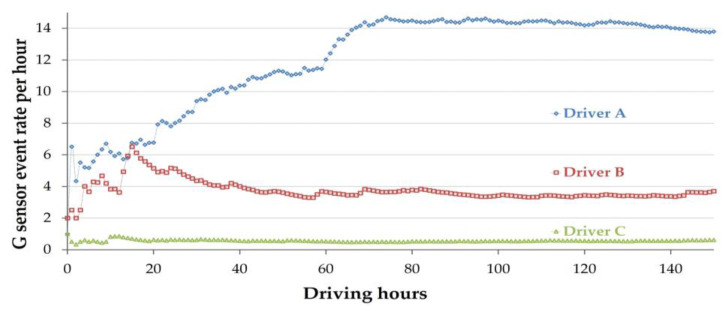
Mean event rate for three drivers (G-sensor data). Driver A’s rate peaks at around 70 h, with a 14 events/hour rate. Driver B starts at a 6.5 events/hour rate, stabilizing to a 3.5 events/hour at 50 h, representing most participants. Driver C stabilizes at a lower 0.5 events/hour rate, below the sample’s average.

**Figure 4 sensors-23-08887-f004:**
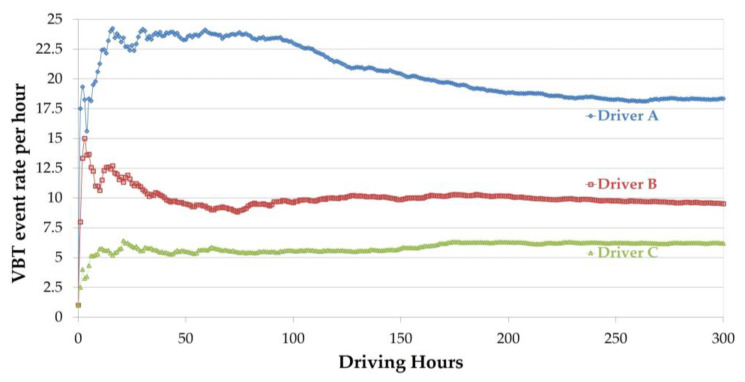
Mean event rate for three drivers (VBT data). Driver A’s rate peaks at ~24 events/hour by 15 h, declines, and stabilizes at 19 events/hour by 100 h. Driver B’s rate stabilizes at 10 events/hour after 75 h, while Driver C’s rate stabilizes at 6 events/hour after around 20 h.

**Figure 5 sensors-23-08887-f005:**
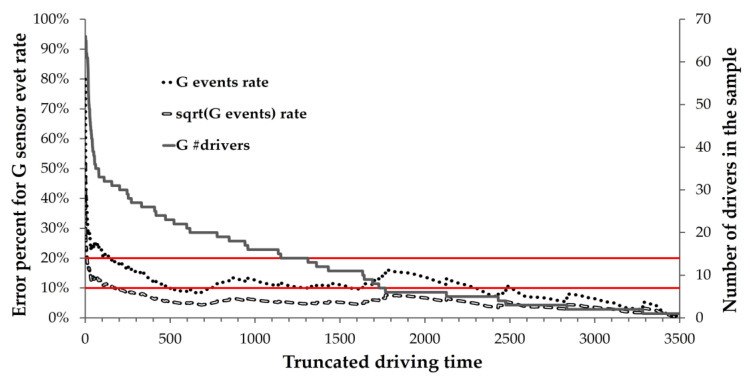
Expected error in G-sensor events based on truncation time. The *x*-axis shows monitored driving hours. The right *y*-axis denotes declining driver counts with increased hours, while the left *y*-axis plots the estimated G-sensors event rate error. The area between the two red lines highlights the 10% to 20% error range. Three lines are featured: the G-sensor event rate (dotted line) and its square root (dashed line) refer to the error percentage (left *y*-axis), and the third line (continuous) refers to the number of drivers (right *y*-axis).

**Figure 6 sensors-23-08887-f006:**
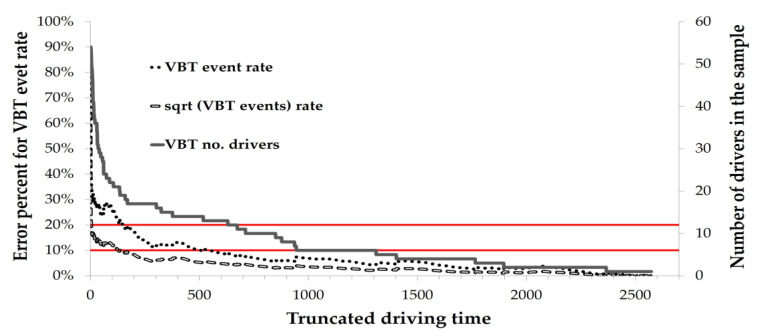
Expected error in VBT events based on truncation time. The *x*-axis shows monitored driving hours. The right *y*-axis denotes declining driver counts with increased hours, while the left *y*-axis plots the estimated VBT event rate error. The area between the two red lines highlights the 10% to 20% error range. Three lines are featured: the VBT event rate (dotted line) and its square root (dashed line) refer to the error percentage (left *y*-axis), and the third line (continuous) refers to the number of drivers (right *y*-axis).

**Table 1 sensors-23-08887-t001:** List of potential events recorded by the VBT system.

Forward Collision Monitoring (FCM)	Urban Forward Collision Monitoring (UFCM)	Unsafe Headway Monitoring (HM)	Unsignaled Lane Deviations (LDs)
Obstacle in front of vehicle is <2.7 s away	Obstacle in front of the vehicle is <2.7 s away	Obstacle in front of the vehicle is <1 s away	Unplanned deviation without signaling
Active at speeds of >30 kph	Active at speed of <30 kph; better adapted for slow speeds/city traffic jams	Active at speeds of >30 kph	Active at speeds of >55 kph or higher; better adapted for long-distance travel

**Table 2 sensors-23-08887-t002:** Means and standard deviations of undesirable G-sensor events.

	Turns/Hr	Accel/Hr	Braking/Hr	Sum of Events/Hr
*n* (no. of drivers)	66	54	66	66
Mean	3.55	0.54	0.78	4.77
Std. deviation	2.55	1.21	1.16	3.85
Minimum	0.37	0.01	0.00	0.45
Maximum	12.16	8.63	8.42	21.03

**Table 3 sensors-23-08887-t003:** Means and standard deviations of undesirable VBT events.

	FCM/Hr	UFCM/Hr	HM/Hr	LDs/Hr	Sum of Events/Hr
*n* (no. of drivers)	61	61	61	59	61
Mean	4.41	8.99	4.90	2.12	20.42
Std. deviation	4.88	6.24	6.81	1.92	15.08
Minimum	0.00	0.00	0.00	0.00	1.83
Maximum	24.31	26.17	30.33	8.00	65.12

FCM = forward collision monitoring, UFCM = urban forward collision monitoring, HM = unsafe headway monitoring (HM), LDs = sudden lane deviations.

## Data Availability

Not applicable.

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
