# Peer review of "Optimal Duration of In-Vehicle Data Recorder Monitoring to Assess Bus Driver Behavior"

_sensors, 2023, doi:10.3390/s23218887_

Round 1
Reviewer 1 Report
Comments and Suggestions for Authors
The paper needs improvement.
1. It needs to provide a literature review section.
2. The introduction section needs to provide a clear research gap.
3. Tables 2 and 3 cannot see clear text.
4. Sections 5 and 6 should be combined. Also, what are the academic and managerial implications?
5. The literature is outdated. Please conduct the recent literature review.
Comments on the Quality of English LanguageThe paper needs to conduct a professional editing.
Reviewer 2 Report
Comments and Suggestions for Authors
In this paper, the Authors are proposing to examine the optimal sampling durations for In-Vehicle Data Recorder (IVDR) data analysis.
A study has been carried out involving professional bus drivers.
The results show that an optimal sampling time frame may be required for achieving a stable and reliable driving pattern for Vision-Based Technology and G-sensors.
I have found the study really interesting and appropriate for this problem.
After carefully reading, I find that this paper is extremely interesting, however in order to further improve I would only recommend to improve the conclusions and more references on the background
Reviewer 3 Report
Comments and Suggestions for Authors
This paper examines the optimal sampling durations for In-Vehicle Data Recorder (IVDR) data analysis, focusing on professional bus drivers. Vision-Based Technology (VBT) from Mobileye Inc. is an emerging technology for monitoring drivers' behavior and enhancing safety in advanced driver assistance systems (ADAS) and autonomous driving. VBT assesses distances to vehicles, detecting hazardous driving events.
My comments are as follows.
1. The motivation is not quite clear for me. What is the main shortcoming for current solutions?
2. “The limited research available on this topic in the literature and the wide range of sampling durations suggested by the existing studies reveals a critical information gap on the ideal monitoring duration to yield accurate driving assessments” How to explain this sentence?
3. The length of this paper is a bit short. More detail of the scheme design should be added and discussed.
4. The figures are not clear. High-level generation tools should be employed to enhance the quality of figures.
5. Some important work, such as smart collaborative evolvement for virtual group creation in customized industrial iot, on developing a driver identification methodology using in-vehicle data recorders, should be added.
6. The proofreading is highly needed.
Comments on the Quality of English LanguageOK
Round 2
Reviewer 1 Report
Comments and Suggestions for Authors
One minor comment is that the last two sections combine will look better.
Author Response
"Please see the attachment."
